# Curcumin and Silver Doping Enhance the Spinnability and Antibacterial Activity of Melt-Electrospun Polybutylene Succinate Fibers

**DOI:** 10.3390/nano12020283

**Published:** 2022-01-17

**Authors:** Maike-Elisa Ostheller, Abdelrahman M. Abdelgawad, Naveen Kumar Balakrishnan, Ahmed H. Hassanin, Robert Groten, Gunnar Seide

**Affiliations:** 1Aachen-Maastricht Institute for Biobased Materials (AMIBM), Maastricht University, Brightlands Chemelot Campus, Urmonderbaan 22, 6167 RD Geleen, The Netherlands; gunnar.seide@maastrichtuniversity.nl; 2Textile Technology Research Institute, National Research Center (Affiliation ID: 60014618), Dokki, Cairo 12622, Egypt; aabdelgawad2@gmail.com or; 3Aachen-Maastricht Institute for Biobased Materials e.V. (AMIBM e.V.), Lutherweg 2, 52068 Aachen, Germany; naveen.balakrishnan@maastrichtuniversity.nl; 4Materials Science & Engineering Department, School of Innovative Design Engineering, Egypt-Japan University of Science and Technology (E-JUST), Alexandria 21934, Egypt; ahmed.hassanin@ejust.edu.eg; 5Department of Textile Engineering, Faculty of Engineering, Alexandria University, Alexandria 21934, Egypt; 6Department of Textile and Clothing Technology, Niederrhein University of Applied Sciences, Campus Moenchengladbach, Webschulstrasse 31, 41065 Moenchengladbach, Germany; Robert.Groten@hs-niederrhein.de

**Keywords:** polybutylene succinate, melt electrospinning, biomedical applications, antibacterial nonwovens

## Abstract

Melt electrospinning is a polymer processing technology for the manufacture of microfibers and nanofibers. Additives are required to reduce the melt viscosity and increase its conductivity in order to minimize the fiber diameter, and can also impart additional beneficial properties. We investigated the preparation of polybutylene succinate (PBS) microfibers incorporating different weight percentages of two multifunctional additives (the organic dye curcumin and inorganic silver nanoparticles) using a single-nozzle laboratory-scale device. We determined the influence of these additives on the polymer melt viscosity, electrical conductivity, degradation profile, thermal behavior, fiber diameter, and antibacterial activity. The formation of a Taylor cone followed by continuous fiber deposition was observed for compounds containing up to 3% (*w/w*) silver nanoparticles and up to 10% (*w/w*) curcumin, the latter achieving the minimum average fiber diameter of 12.57 µm. Both additives reduced the viscosity and increased the electrical conductivity of the PBS melt, and also retained their specific antibacterial properties when compounded and spun into fibers. This is the first report describing the effect of curcumin and silver nanoparticles on the properties of PBS fibers manufactured using a single-nozzle melt-electrospinning device. Our results provide the basis to develop environmentally benign antibacterial melt-electrospun PBS fibers for biomedical applications.

## 1. Introduction

Electrospinning is a cost-effective, versatile and efficient method for the preparation of fibers in the low micrometer to nanometer range [1,2,3]. These highly flexible fibers feature an enormous surface area, making them suitable for applications in biomedicine [4], filtration and separation [5], textile manufacturing, electronics and energy [6,7]. Electrospinning involves the creation of an electric field by establishing a potential difference between the end of a needle capillary and a collector [8]. This confers a surface charge upon a polymer solution or melt, resulting in the formation of a so-called Taylor cone. When the electrostatic repulsive force of the surface charge overcomes a specific surface tension, a charged jet stream is ejected from the tip of the Taylor cone. The charge density of the jet interacts with the external field and causes whip-like instabilities, which stretch the fiber and reduce its diameter to the micrometer/sub-micrometer range. The continuous filaments are deposited as a nonwoven-like material on the collector [9].

Melt electrospinning is an environmentally benign and solvent-free alternative to solution electrospinning, but some challenges must be overcome before this method becomes economically feasible at scale [10,11,12,13,14]. One of the key issues is the processability of the polymer melt, which has a much higher viscosity and lower conductivity than polymer solutions [15]. It is also important to reduce the utilization of petrochemical polymers by replacing them with biobased and biodegradable thermoplastic alternatives such as polybutylene succinate (PBS) [16].

We previously reported the first melt-electrospun PBS fibers in the low micrometer range prepared using a laboratory-scale device [17]. The minimum fiber diameter was 30.05 µm, significantly thinner than fibers produced under similar conditions using other biopolymers such as polylactic acid [4]. However, even the finest melt-electrospun fibers are thicker than fibers produced by solution electrospinning. Given the high cost of machine optimization, additives are therefore required to overcome viscosity and conductivity limitations of molten PBS and enhance its processability to reduce the fiber diameter. 

Typical additives used in melt-electrospinning include salts and plasticizers. However, dye pigments and metallic nanoparticles are also suitable because of their positive impact on conductivity [18]. If such additives also impart further useful properties, they are known as multifunctional additives. For example, the natural dye curcumin, silver nanoparticles and graphene oxide silver nanofillers are both well-known as antibacterial agents [19,20,21]. Curcumin, formally known as 1,7-bis-(4-hydroxy-3-methoxyphenyl)-1,6-heptadiene-3,5-dione, is naturally produced by the rhizomes of *Curcuma longa* L. (Zingiberaceae) and is the principal active constituent of the spice turmeric. It is active against a wide range of microorganisms [22]. The mechanism of action involves the absorption of light (300–500 nm, with a maximum absorption band at 425 nm) allowing it to act as a photosensitizer [23]. In contrast, silver nanoparticles are thought to attach to bacterial cell walls and interfere with permeability and the exchange of nutrients. Some particles may also penetrate the cell and interact with proteins and nucleic acids to disrupt DNA replication. Silver ions released from the nanoparticles may also be toxic to the bacteria [24]. Silver compounds have, therefore, been incorporated into materials used as medical dressings to prevent microbial infections [25]. Graphene oxide-silver nanoparticles have also proven to be very effective against *E**. coli* and *Staphylococcus aureus* on LB-agar plates and have been successfully tested as nanofillers in membranes [21]. However, to the best of our knowledge, these additives have been mostly tested in solution processing methods. 

Here, we investigated the effect of curcumin and silver nanoparticles on the viscosity and conductivity of PBS during melt electrospinning in order to minimize the fiber diameter. Neither additive has previously been used to tailor a PBS melt-electrospinning process, so this is the first report describing the properties of such fibers prepared using a single-nozzle melt-electrospinning device. Furthermore, we also tested the antibacterial activity of the PBS-compound fibers to determine their suitability for biomedical applications. We tested fibers incorporating different weight percentages of each additive and determined their influence on the polymer melt viscosity, electrical conductivity, degradation profile, thermal behavior, fiber diameter, and antibacterial activity. Our results can be used as the basis to develop environmentally benign melt-electrospinning processes that use multifunctional additives to manufacture microscale or nanoscale fibers for biomedical applications.

## 2. Materials and Methods

### 2.1. Materials

PBS fiber-grade resin (FZ78TM) produced by the polymerization of biobased succinic acid and 1,4-butanediol was used as the base polymer for all experiments. The following specifications were reported by the manufacturer (MCPP, Düsseldorf, Germany): melt flow rate = 22 g/10 min at 190 °C using a weight of 2.16 kg, and crystalline melting temperature = 115 °C. Curcumin from turmeric powder, soybean protein, and silver nitrate were sourced from Sigma-Aldrich (Zwijndrecht, Netherlands). The chemical structures and melting points of PBS [26] and the curcumin [27] are shown in Table 1.

### 2.2. Micro-Compounder

PBS and curcumin powder were dried at 60 °C under vacuum for 12 h before processing. Silver was dried at 240 °C for 1 h before processing to promote calcination (see Section 3.2). PBS compounds were prepared by mixing 1%, 3%, 5%, 7% or 10% (*w/w*) of each additive in a micro-compounder (Xplore, Sittard, Netherlands) at 200 °C with a screw speed of 100 rpm for 2 min. The compounds and their abbreviations are summarized in Table 2. 

### 2.3. Methods

#### 2.3.1. Synthesis of Silver Nanoparticles

Silver nanoparticles were prepared using soybean protein, as we previously described by us [28]. Briefly, soybean protein was employed as a reducing and capping agent via solid-state reaction. First, the globular structure of soybean protein (5 gm dry powder) was activated by adding sodium hydroxide (0.05 gm). Then, the two compounds were mixed thoroughly using a manual mortar for 10 min.

Exactly 1 gm silver nitrate (AgNO_3_) was added to the mixture, and vagarious mixing was continued for an additional 15 min at the same temperature. As a result, soybean protein-stabilized AgNPs were formed. The resultant solid mixture exhibited a deep brown color. The as-prepared powder was kept in a dry container for further investigation and characterization.

#### 2.3.2. Melt-Electrospinning Equipment

A single-fiber melt-electrospinning device consisting of a temperature controller, high-voltage power supply, heating elements, syringe, pump and collector was used to evaluate the general processability and fiber formation characteristics of PBS compounds. The device was equipped with JCS-33A temperature process controllers (Shinko Technos, Osaka, Japan) and PT 100 platinum thermocouples (Omega Engineering, Deckenpfron, Germany) to control the melting temperature (Figure 1).

The trials with all polymer compounds were carried out at 235 °C. We used a KNH65 high-voltage generator (Eltex-Elektrostatik, Weil am Rhein, Germany) with a 6–60 kV range, which was set to a constant 60 kV during all melt-electrospinning experiments (positive voltage on the collector and a grounded nozzle-tip). A flat aluminum plate (6 cm diameter) overlaid with thin paperboard was used as the collector. During the melt electrospinning experiments, the nozzle-to-collector distance was kept constant at 10 cm. An 11 Plus spin pump (Harvard Apparatus, Cambridge, MA, USA) was used with a feed rate of 0.1 mL/min. A 2-mL glass syringe (Poulten & Graf, Wertheim, Germany) equipped with an additional metal orifice of 0.90 mm served as the spinneret nozzle. 

#### 2.3.3. Characterization of Materials and Compounds

UV-Vis absorption spectra (wavelength from 800 to 300 nm) of the suspended silver nanoparticles were attained using a Shimadzu (UV1601) spectrophotometer at given time intervals. Surface plasmon resonance (SPR) is the most characteristic property to demonstrate the formation of metal nanoparticles.

The particle morphology, size diameter of the silver nanoparticles was retrieved via transmission electron microscopy (TEM, JEOL JEM 2010) operating at an accelerating voltage of 90 kV. The silver nanoparticles in the solid-state were diluted with deionized water, and the dispersion was subjected to sonication for 15 min. Then, one drop was placed on a carbon-coated copper grid and allowed to dry for 5 min prior to TEM imaging.

Differential scanning calorimetry (DSC) was carried out using a Q2000 device (TA Instruments, Asse, Belgium). We focused on changes to the melting temperature (Tm), recrystallization temperature (Trc), crystallization temperature (Tc), and crystallinity (Xc) caused by different weight ratios of additives from the second cycle. All PBS compounds were tested at a heating rate of 10 °C/min, using a temperature range of –60 to +200 °C with a sample size of ~5 mg. The percentage crystallinity (Xc) of PBS and its compounds was calculated using Equation (1): (1)$$X_{c}\left(\%\right)=\frac{\Delta Hm}{\Delta Hm^{0}*\left(\mathrm{wt}.\%\right)}*100$$
where *Xc* is the percentage crystallinity, ΔHm is the melting enthalpy of PBS compounds, $\Delta Hm^{0}$ is the melting enthalpy of 100% crystalline PBS, and wt.% is the weight percentage of the additive. The melt enthalpy of 100% crystalline PBS was considered to be 110.3 J/g [29]. All compounds were prepared using the same protocol, so any differences in material properties should primarily reflect the nature and quantity of additives.

Thermogravimetric analysis (TGA) was carried out using a Q5000 device (TA Instruments). We heated 5 mg of each PBS compound at a rate of 10 °C/min under nitrogen flowing at 50 mL/min until the temperature reached 700 °C. We also conducted an isothermal study at a constant temperature of 235 °C (spinning temperature) for 60 min to observe any weight loss over time. The temperatures at which 5% and 50% weight loss occurred were determined using TA universal analysis software, and the values were compared to determine the influence of additives on the thermal stability of PBS.

The rheological properties of PBS compounds were characterized using a Discovery HR1 hybrid rheometer (TA Instruments, New Castle, DE, USA) focusing on the frequency-dependent complex viscosity G*. We carried out a frequency sweep from 1 to 628 rad/s. For all experiments, we used a 25-mm plate-to-plate geometry with the distance set to 1000 μm, and the strain amplitude was maintained at 1%. Measurements were taken at a temperature of 235 °C. The complex viscosity of pure PBS and PBS compounds is presented at an angular frequency of 10 rad/s and at a temperature of 235 °C. 

We also measured degradation in terms of molecular weight (after compounding and melt electrospinning) by gel permeation chromatography (GPC) using a 1260 Infinity System (Agilent Technologies, Santa Clara, CA, USA). Hexafluor-2-isopropanol (HFIP) containing 0.19% sodium trifluoroacetate was used as the mobile phase at a flow rate of 0.33 mL/min. Solutions were prepared by dissolving 5 mg of pure PBS and the various compounds in HFIP for ~2 h, passing the solutions through a 0.2-µm polytetrafluoroethylene filter, and injecting them into a modified silica column filled with 7-µm particles (Polymer Standards Service, Mainz, Germany). The experiment was calibrated against a standard polymethyl methacrylate polymer (1.0 × 10^5^ g/mol), and the relative weight average molecular weight (M_w_), number average molar mass (M_n_), and polydispersity index (PDI) of each polymer were recorded and compared.

The electrical resistance of the molten PBS compounds was measured at the same temperatures as the rheological properties using a Keithley 617 electrometer (Tektronix, Beaverton, OR, USA). The polymer granules were melted using band heaters, and two electrodes (6 mm apart) were dipped into the melt and connected to the electrometer. The current flowing between the electrodes was measured at 10 V (Figure 2).

Fourier transform infrared (FTIR) spectrophotometry was carried out using an IRAffinity-1 instrument (Shimadzu Deutschland, Duisburg, Germany) taking 64 scans per measurement at a wavelength ranging from 550–3850 nm. We performed the scans in absorption mode with a spatial resolution of 2 cm^−1^.

#### 2.3.4. Characterization of Fibers

Fiber diameters were measured using an Olympus BX53 microscope fitted with an Olympus DP26 camera (Olympus, Leiderdorp, Netherlands). The fiber diameter was measured 100 times at different positions for each sample based on the 50× magnified image.

The effect of melt electrospinning on the molecular weight of PBS compound fibers was determined by GPC as described above. Fiber morphology was investigated by field emission scanning electron microscopy (FE-SEM) using a JSM-6400 instrument (JEOL, Peabody, MA, USA) operating at 10–20 kV with a working distance of 20 mm. Before analysis, a small piece of the fiber web was fixed on carbon tape and then sputtered with a thin gold layer for 3 min.

#### 2.3.5. Biological Activity

The antibacterial activity of the additives and fibers was tested against *Escherichia coli* ATCC 25922 (G−ve) and *Staphylococcus aureus* ATCC 6538 (G+ve) using the viable cell count method, by assessing the number of colony-forming units (CFUs) [24]. For screening the activity of silver particles and curcumin, an overnight culture of each bacterial strain was prepared. The experiment started by inoculating five tubes filled with 10 mL of nutrient broth liquid medium with 50 μL of bacterial stock (7.4 × 10^8^ CFU). We then added 100 μL of curcumin or silver nanoparticles (2, 5, 10, 20 or 40 mg/5mL distilled water), respectively, to the tubes and incubated for 24 h at 37 °C, shaking at 250 rpm. Untreated cultures were used as controls. The cultures were then serially diluted to obtain bacterial concentration (7.4 × 10^4^ CFU), and 100-μL aliquots were plated on a nutrient agar solid medium. After incubation for 24 h of at 37 °C, the number of colonies was counted, and the reduction rate (R) was calculated in relation to the untreated control using Equation (2).
R (%) = [(B − A)/B] × 100(2)
where, A and B are the CFU/mL values for treated and untreated samples, respectively, after 24 h [30]. The experiments were done in triplicates. 

To test the antibacterial efficiency of the functionalized fibers, the sliver-loaded fibers were tested against *E. coli*, while curcumin-loaded ones against *S. aureus*. Separately, both bacterial strains (100 μL stock of 7.4 × 10^7^ CFU) were inoculated in a 10 mL of freshly prepared liquid nutrient (every 100 mL contains peptone 5 g/L, beef extract 3 g/L) for 24 h. In inoculation flasks (with 20 μL of inoculums), fiber samples have been inserted, leaving the control (sample-less inoculated flasks). A serial dilution of each cultivation containing a sample and control (10^−1^–10^−5^) was performed after 16 h of incubation at 37 °C. A 100 μL from each dilution was spread onto an agar plate, and the reduction rate was calculated. The experiments were done in triplicates.

## 3. Results and Discussion

The Results and Discussion section is divided into 3 parts. Section 3.1 deals with the material analysis and shows to what extent the individual materials are stable under the processing conditions. The following Section 3.2 examines the polymer compounds with special attention to degradation. The last part of the Results and Discussion Section 3.3 is the fiber analysis. Since compounding takes place at a lower temperature than melt electrospinning, possible degradation due to the higher temperature as well as repeated thermal processing of the compounds is again investigated. In addition, it is examined whether the effect of both additives is affected by the repeated heat processing. It will also be investigated whether the final fiber diameter of PBS is affected by the additives.

### 3.1. Material Analysis

The results obtained from the raw material analysis are discussed in this subsection.

#### 3.1.1. Characterization of Silver Nanoparticles

The TEM micrograph of silver nanoparticles prepared using soybean protein as a reducing and capping agent are shown below in Figure 3a–c. It is clear from the figure that the particles have a spherical shape, and no aggregation is noticed. The particle size distribution (Figure 3b) was measured using image-J software, and the average particle size was found to be (20nm). Moreover, the characteristic surface plasmon absorbance peak of silver nanoparticles is observed at 426 nm on the UV-Vis spectrophotometer graph (Figure 3c). The peak appears narrow, which suggests the particles’ low polydispersity, which is also supported by the TEM measurement. 

#### 3.1.2. Thermal Degradation of Materials Used

The decomposition temperatures of pure PBS and the pure additives were determined by TGA. Table 3 shows the residual weights before and after the isothermal step and the 50% weight loss temperatures for each material.

TGA revealed that all the materials undergo single-step degradation (Figure 4a). To assess the thermal stability of the materials, we monitored their isothermal decomposition at 235 °C for 60 min (Figure 4b). During the isothermal step, the weight loss of the naïve silver nanoparticles was 13.14% (from 89.67% to 76.53%). The naïve particles consist of 46.3% silver and 53.7% organic components containing carbon, oxygen, sodium and nitrogen [28]. It is likely that volatiles such as CO_2_ are released during thermal processing, negatively affecting the spinnability of the compounds. We therefore included a calcination step to eliminate any elements that might interfere with spinning. We expected to remove the organic components that degrade at this temperature and can cause a negative impact on the melt electrospinning process. However, we do not expect the organic components, that are stable at this temperature to be removed during this step. After calcination, the weight loss of the silver nanoparticles during the isothermal step was reduced to 0.61% (from 97.58% to 96.97%), which is unlikely to have a negative impact during melt electrospinning. We also observed a weight loss of 1.74% for PBS and 8.14% for curcumin. The weight loss from PBS is negligible, but the weight loss from curcumin has the potential to cause problems during compound processing. However, the isothermal study lasted 60 min whereas the laboratory-scale melt electrospinning machine processes the compounds at 235 °C for only ~5 min. We therefore anticipate no impairment of the process in the small-scale device, but further investigation is required to ensure an effective transfer to the pilot-scale machine, which has a dwell time of ~45 min.

The 50% weight loss temperature of pure PBS was 398 °C. The corresponding values for curcumin and silver nanoparticles before calcination were lower (389 °C and 383 °C, respectively), but the value for the calcinated silver nanoparticles was outside the range we tested and is therefore not reported. The 50% weight loss temperatures of all components were significantly higher than the process temperature of 235 °C, suggesting that the materials are unlikely to experience significant degradation during melt electrospinning.

#### 3.1.3. Thermal Properties of the Materials

The DSC thermograms of pure PBS, curcumin and silver nanoparticles are presented in Figure 5. The DSC thermogram of the PBS granules featured two melting peaks. A small peak was observed at ~50 °C, which may reflect the presence of a plasticizer used to improve the processability of PBS during manufacturing or (given that we used undried granules) may indicate the evaporation of water. A second peak at ~110 °C corresponds to the melting point reported by the manufacturer. The DSC thermogram of curcumin featured a single melting peak at 175 °C. The observed values are in agreement with the previously reported values [17]. The DSC thermogram of silver extract indicated a melting range rather than a clear melting point, which does not belong to the silver particle, but may reflect removal of water present in the extract and the high polydispersity of the organic components present within the silver nanoparticles and the presence of a wide range of particle sizes. Small clusters of particles melt at lower temperatures than larger ones [31]. The small melting peak observed near 215 °C could belong to the residual silver nitrate left in the material after synthesis. The cooling cycle of the DSC thermogram reveals a crystallization peak at ~60 °C for pure PBS, whereas the additives showed crystallization over the whole temperature range. 

#### 3.1.4. Antibacterial Properties of the Active Materials

The antibacterial activity of curcumin and silver nanoparticles was evaluated against *E. coli* and *S. aureus* using the viable cell counting method (Figure 6). The number of colonies was recorded after incubating a fixed number of bacterial cells with different concentrations of active materials for 24 h. Curcumin showed bactericidal activity against the Gram-positive *S. aureus* strain at a concentration of 15 mg/mL, whereas the silver nanoparticles were only bacteriostatic against *S. aureus* at 100 mg/mL. However, the silver nanoparticles showed bactericidal activity against the Gram-negative *E. coli* strain at a concentration of 5 mg/mL, whereas curcumin was only bacteriostatic against *E. coli* at 100 mg/mL. The ability of Gram-negative bacteria to withstand curcumin reflects the structure of their cell walls [22]. These observations are consistent with previous studies in which curcumin showed phototoxicity toward planktonic *S. aureus* and *S. epidermidis* [32,33] and silver ions were toxic toward *E. coli* [34].

### 3.2. Compound Analysis

The results obtained from the compound analysis are discussed in this subsection.

#### 3.2.1. Thermal Analysis of the Compounds

The decomposition temperatures of pure PBS and the PBS-curcumin and PBS-silver compounds were determined by TGA. 

All compounds underwent single-step degradation (Figure 7). During the isothermal step, all PBS-curcumin compounds showed a small weight loss of ~1% and all PBS-silver compounds similarly lost ~2–3% of the original weight, regardless of the weight ratio of the additive. PBS showed a linear decrease in weight over time, falling to 97.75% after 60 min at 235 °C. Compound S1 showed the highest weight loss during the isothermal step (decreasing to 96.61% after 60 min), whereas compound S7 showed the lowest weight loss (decreasing to 98.11% after 60 min). These weight losses are not significant for a manufacturing process, and as discussed above, the exposure to high temperatures in the laboratory-scale device is ~5 min, suggesting that the effects will be negligible. However, as stated for the pure materials, the thermal degradation behavior may need to be reinvestigated for pilot-scale manufacturing due to the longer dwell time.

Table 4 shows the residual weights before and after the isothermal step, and the 50% weight loss temperatures for each compound. 

The 50% weight loss temperature differed significantly among the compounds. For all compounds except S10, the 50% weight loss temperature was higher than that of pure PBS. Compound C7 showed the highest 50% weight loss temperature of 366.75 °C, whereas the lowest value was recorded for S10 at 352.66 °C. PBS was previously shown to undergo single-step degradation with 50% weight loss at 398 °C. However, we applied an isothermal step at 235 °C that was not included in previous studies. We observed a slight weight loss during this isothermal step, and because degradation started earlier, the degradation shifted to lower temperatures and the 50% weight loss temperature was correspondingly lower. The 50% weight temperatures for the PBS-silver compounds ranged from 352 °C (S10) to 358 °C (S1). Both of the additives therefore stabilized the thermal behavior of PBS. The 50% weight loss temperatures of all compounds were significantly above the process temperature of 235 °C and are therefore not expected to interfere with processing. As TGA can only measure weight loss in the form of released volatile gases, we also characterized degradation leading to molecular weight reduction and oligomer formation by GPC and rheological analysis.

#### 3.2.2. Thermal Properties of the Compounds

The DSC thermograms of PBS and its compounds are shown in Figure 8. The Tm, Trc, Tc, and Xc values are summarized in Table 5. The DSC thermograms of pure PBS and its compounds featured a melting peak (Tm2) at ~110 °C, which is consistent with the PBS melting point reported by the manufacturer. Pure PBS and the PBS-curcumin-compounds also featured an additional shoulder (a secondary melting peak) at ~105 °C. This may reflect the presence of a secondary crystal structure or the formation of crystals of different sizes, but further investigation is required to define the precise cause. We also observed a broadening of the melting peak as the curcumin concentration increased, which may also be caused by the presence of crystals over a range of different sizes. PBS and all its compounds featured another small melting peak at ~50 °C, which as stated above may indicate the presence of a plasticizer introduced during the manufacture of PBS pellets [17]. The DSC thermogram of PBS and its compounds showed a recrystallization peak at ~107 °C (Trc). During cooling, the crystals that form may differ in size or type and may contain various defects. As the PBS is heated, these small crystals (and in some cases the crystal defects) can melt and recrystallize or combine to form larger crystals, which melt again at the higher temperature. The recrystallization peak could also reflect the presence of amorphous polymer chains that become more mobile at higher temperatures and form a more ordered crystalline structure, as previously reported [35]. The DSC results showed that the Tm2 of PBS is not markedly affected by the nature or concentration of the additives, remaining at ~112 °C. In contrast, we observed a significant additive-dependent difference in Tc. The Tc of pure PBS was 74 °C, but this declined in a linear manner as the concentration of curcumin increased, reaching a minimum value of 68 °C (C10). Increasing the concentration of silver also slightly reduced the Tc, reaching a minimum value of 72 °C (S10). Curcumin and silver therefore appear to slow down the crystallization of PBS. 

The Xc of the PBS-curcumin compounds was slightly higher than that of pure PBS. Although the addition of curcumin delays the crystallization process, crystallinity nevertheless increases marginally at higher weight percentages. As stated above, this is accompanied by a broadening of the melting peak, which also indicates the broad distribution of polymer crystals and crystal sizes. The slight increase in crystallinity could therefore reflect the formation of crystals varying in type and size. In contrast to curcumin, silver did not appear to influence the Xc, which remained at ~55% even at the highest silver concentration (Table 5).

#### 3.2.3. Molecular Weight of the Compounds

The GPC curves of pure PBS and its compounds are presented in Figure 9.

The relative Mw of extruded PBS was 85,710 Da, the Mn was 37,970 Da, and the PDI was 2.02. These values did not change significantly in the presence of additives (Figure 10). The lowest Mw and Mn values were observed for compound S10 (79,990 and 32,600 Da, respectively) and the highest were observed for compound C5 (89,710 and 41,860 Da, respectively) (Figure 10). The PDI of all compounds fell within the range 2.14–2.47. The changes in molecular weight showed no clear trend and were in any case within the experimental error range accepted for GPC. We therefore concluded that none of the additives induce the degradation of PBS. 

#### 3.2.4. Effect of the Additives on Viscosity

The complex viscosity of pure PBS and its compounds was plotted as a function of angular frequency at different weight ratios of each additive at a set temperature of 235 °C (Figure 11). 

Pure PBS showed a complex viscosity of 131.56 Pa·s, but this could be reduced by the addition of either curcumin or silver. Among the PBS-curcumin compounds, C7 showed the lowest melt viscosity (54 Pa·s, ~41% of the pure PBS value). The addition of curcumin therefore has a plasticizing effect, but the higher complex viscosity of C10 compared to C7 indicates that the effect is not solely determined by the weight ratio (Figure 12a). The addition of silver had a much more significant effect on viscosity than curcumin. The compound with the lowest complex viscosity was S3 (19.88 Pa·s, ~15% of the pure PBS value). This is in the middle of the weight ratio range, again showing that the weight ratio is not the sole determinant of complex viscosity (Figure 12b). The behavior of the compounds could be caused by polymer degradation, although this has been ruled out by GPC analysis (Section 3.2.3). We therefore conclude that both additives act as plasticizers. In the case of curcumin, the melting point is 175 °C and curcumin is therefore molten under our testing conditions. Similarly, the organic residuals of the capping agent (soybean protein isolate) in the silver extract have also low molecular weight, which leads to lower viscosity. The interaction between the polar groups of PBS and (a) polar groups in curcumin or (b) organic residuals in the silver extract could lead to the swelling of the polymer chains, thus increasing the free volume. Such interactions would reduce intermolecular cohesion and increase polymer chain mobility, thus reducing the viscosity of the polymer melt [36]. 

The lower complex viscosity of the compounds compared to PBS could improve the spinnability of the melt because lower viscosity facilitates jet thinning during the spinning process and therefore leads to the production of narrower fibers. However, if the viscosity is too low, the melt could flow too rapidly, with insufficient time for the whipping and elongation of the jet. We included an isothermal step during which the shear was kept constant, allowing us to determine the effect of time on viscosity (Figure 13).

For all PBS-curcumin compounds, the viscosity was high initially, but fell progressively as the step time increased. This differed from the rheological behavior of pure PBS, where the viscosity increased over time. The greatest decline in viscosity was observed for compound C7. The behavior of the PBS-curcumin compounds probably reflects the degradation of curcumin at prolonged high temperatures as previously reported [37], which we propose may lead to the accumulation of byproducts that promote the degradation of PBS. In contrast, the increasing viscosity of pure PBS over time may indicate the formation of cross-links or branches in the polymer. Unlike the PBS-curcumin compounds, which showed similar viscosity time curves, the behavior of the PBS-silver compounds differed according to the weight percentage. An initial increase in the complex viscosity of compound S1 was observed at a constant angular frequency over 60 min whereas the other PBS-silver compounds showed a decrease in viscosity under the same conditions. 

#### 3.2.5. Effect of Additives on Electrical Conductivity

The electrical resistance of pure PBS and its compounds was measured at 235 °C (Figure 14).

Electrical conductivity requires freely movable charge carriers. The electrical resistance of pure PBS (75 MΩ) decreased by 40% for C1, 48% for C3, 78% for C5, 80% for C7 and 88% for C10, indicating an inverse relationship between the curcumin content and electrical resistance. This translates to an increase in conductivity for compounds containing a higher weight percentage of curcumin. Similarly, the introduction of curcumin into a thin thiourea film resulted in an increase in electrical conductivity, apparently by generating extensively conjugated double bonds [38]. We assume that the same phenomenon occurs in the PBS-curcumin compounds, further suggesting that compounds with the highest weight ratio may interact to a greater extent with the electrical field during melt electrospinning and possibly generate thinner fibers. A similar phenomenon was observed for the silver compounds, where the presence of the additive reduced the electrical resistance by a factor of five. The electrical resistance decreased significantly even at the lowest weight ratio of silver (11 MΩ for S1), but remained constant at 10 ± 2 MΩ for the other compounds, showing that higher concentrations of silver had no significant effect on electrical resistance.

#### 3.2.6. FTIR Spectroscopy

FTIR spectra characterizing the interactions between PBS and the additives are shown in Figure 15. 

In the PBS trace, the peak at $917{\;\mathrm{cm}}^{-1}$ indicates the -C-OH- bond in the carboxylic acid group, the bands at 1044–1046$\;{\mathrm{cm}}^{-1}$ represent the -O-C-C- stretching vibrations, and the peaks at 1144–1264$\;{\mathrm{cm}}^{-1}$ represent the stretching of the -C-O-C- group in the ester linkages. The peaks in the region of 1710–1713${\;\mathrm{cm}}^{-1}$ correspond to the C=O stretching vibrations in the ester groups and the band at 1330–2945${\;\mathrm{cm}}^{-1}$ can be attributed to the symmetric and asymmetric deformational vibrations of –CH_2_- groups in the PBS main chains [39]. These peaks did not shift in the PBS-silver compounds, and the intensity remained at the same magnitude with the exception of S10. These results suggest that silver does not interact with the PBS polymer chains. In contrast, we observed peak shifts and differences in intensity for compounds C7 and C10, whereas C1, C3 and C5 were unchanged. One potential explanation is the degree of mixing in the curcumin compounds [40]. The FTIR measurement was performed at a random point in each compound and a region 1 µm in length was scanned. Given that curcumin particles exceed this size, it is possible that FTIR measurements by chance focused on an area containing such particles, resulting in spectra more closely related to the pure curcumin trace (Figure 15a). This would indicate that, even at high weight ratios of curcumin, there may be no chemical interactions between PBS and the additive. The interactions between PBS and both curcumin and silver may therefore be entirely physical.

### 3.3. Fiber Analysis

The results obtained from the fiber analysis are discussed in this subsection.

#### 3.3.1. Fiber Diameter and Distribution

The melt electrospinning of PBS and its compounds at 235 °C was successful. Taylor cone formation followed by typical fiber deposition was observed for PBS-silver compounds at weight ratios of 1% and 3%, and for PBS-curcumin compounds at all weight ratios. The average diameter of the pure PBS fiber was 30.05 µm, as determined by light microscopy. All compounds produced narrower fibers than pure PBS (Figure 16). 

As stated above, thinner fibers can be produced by melt electrospinning if the viscosity of the melt is reduced and/or the electrical conductivity is increased. Both additives reduced the viscosity of the melt, but the viscosity was not directly proportional to the additive weight ratio (Figure 11). Furthermore, the thinnest fibers were produced by compound C10 (12.57 µm, 58.2% finer than pure PBS fibers) even though the viscosity of compound C7 was lower. However, compound C10 achieved the highest electrical conductivity of the PBS-curcumin compounds (10 MΩ) and there was a proportional albeit nonlinear relationship between the weight ratio of curcumin and the electrical conductivity of the melt (Figure 13). Given that the fiber diameter declines in a linear manner relative to the weight ratio of curcumin (Figure 16), our results suggest that the diameter of PBS-curcumin compound fibers is governed by a combination of increasing conductivity and decreasing viscosity.

Compounds containing 1% or 3% silver produced fibers but higher weight ratios did not. Given that the increased conductivity of the PBS-silver compounds was a uniform property that did not depend on the weight ratio (Figure 13), the inability to spin fibers from compounds S5, S7 and S10 may reflect the presence of higher concentrations of impurities such as ash or byproducts from the calcination step, leading to the formation of agglomerates within the polymer and resulting in particles too large to be spun. This could be addressed in the future by including a filtration step to eliminate such agglomerates. The alternative possibility that S5, S7 and S10 are degraded during melt electrospinning is addressed in Section 3.3.2.

Given that the conductivity of the melt increased significantly even in the presence of the lowest weight ratio of silver, compounds S1 and S3 both produced narrower fibers than pure PBS (20.13 µm for S1 and 16.07 µm for S3, compared to 30.05 µm for pure PBS). Silver achieved a lower melt viscosity and higher conductivity than the same weight ratio of curcumin, resulting in thinner fibers for compounds containing the same proportion of additives.

#### 3.3.2. Analysis of Fiber Morphology by SEM

Figure 17 shows SEM images of PBS fibers containing different amounts of each additive. The images confirm a homogenous distribution of uniform microfibers reflecting the optimized blending and spinning conditions. The fiber diameters measured by SEM agreed with those determined by light microscopy: for example, the pure PBS fibers were ~35 µm in diameter, whereas those containing 1% and 3% silver were ~20 and ~15 µm, respectively, and those containing 1% and 3% curcumin were 23 and 20 µm, respectively. All fibers regardless of composition also featured a circular cross section, but the melting process also generated some fused fibers, a phenomenon that is not observed in solution electrospinning. The homogeneous distribution of additives was also confirmed in the fibers containing silver (Figure 18) and curcumin (Figure 19), highlighting the efficiency of the compounding process. However, it is observed from the images that the particle sizes of the additives are still in the micrometer range. As an optimization, a grinding step can be included to further reduce the size of the additives. This reduction in particle size can lead to higher surface area of the additives which in turn leads to more interaction with the polymer matrix and therefore can potentially improve the functionality of the additives leading to fibers with lower diameter [41]. 

#### 3.3.3. Analysis of Fiber Properties by GPC

The GPC curves of fibers composed of pure PBS and its compounds are presented in Figure 20. The relative Mw of the pure PBS fiber was 67,450 Da, the Mn was 36,610 Da, and the PDI was 1.84. The addition of silver reduced the Mn and Mw in a concentration-dependent manner (S1, Mn = 28,170, Mw = 56,310 and PDI = 2.0; S3, Mn = 25,520, Mw = 50,590 and PDI = 1.98) as shown in Figure 21. Although the GPC analysis of the (unspun) PBS-silver compounds revealed no evidence of degradation, the lower Mn and Mw values of the fibers suggest that silver might induce the degradation of PBS during melt electrospinning. This is possible because compounding was carried out at 200 °C whereas melt electrospinning was carried out at the higher temperature of 235 °C. The degradation of PBS could be induced by the higher processing temperature and/or may simply reflect the fact that melt electrospinning is the second thermal processing treatment to which the compound has been exposed. The GPC results confirm that thermal degradation is clearly one of the reasons for the lower spinnability of PBS-silver compounds.

The relative Mw and Mn did not change significantly for any of the PBS-curcumin fibers compared to the unspun PSB-curcumin compounds and pure PBS (Figure 21). The lowest Mw and Mn values were observed for C5 (61,960 and 29,950 Da, respectively) and the highest were observed for C10 (66,870 and 32,530 Da, respectively). The absence of a significant change in any of the compounds and derived fibers confirms that curcumin does not induce the degradation of PBS during compounding or melt electrospinning.

#### 3.3.4. Antibacterial Activity of the Loaded Fibers

The antibacterial activity of the fibers was tested to determine whether the properties of the pure additives were preserved after compounding and melt electrospinning. Fibers loaded with 3% curcumin were therefore tested against *S. aureus* and those loaded with 3% silver nanoparticles were tested against *E. coli* (Figure 22). No colonies were detected when *E. coli* cultures diluted 10^−5^ were exposed to the silver-loaded fibers, representing a 100% efficiency of reduction. A few *S. aureus* colonies still formed when exposed to curcumin under the same conditions, representing a 98% efficiency of reduction. PVP-capped silver particles were tested as antimicrobial additives in solution electrospun PBS fiber mats in the past and it was reported that the mats had a very high efficiency of 99% against *E. coli*. This is in agreement with the results of our experiments [42]. The active ingredients therefore retained their antibacterial properties despite the high temperatures required for compounding and melt electrospinning, suggesting that fibers containing such additives could be suitable for biomedical applications. 

## 4. Conclusions

We successfully tested the organic dye curcumin and inorganic silver nanoparticles as multifunctional additives to reduce the melt viscosity and increase the conductivity of molten PBS (thus producing thinner fibers during melt electrospinning) while conferring antibacterial properties on the resulting fibers. The compounded fibers were thinner than pure PBS and retained the antibacterial properties of the pure additives despite the high temperature of the compounding and melt electrospinning processes, making the fibers suitable for biomedical applications such as the preparation of wound dressings. The silver-loaded fibers achieved 100% reduction efficiency (no visible colonies) when added to a suspension of *E. coli* diluted to 10^−5^, whereas the curcumin-loaded fibers achieved 98% reduction efficiency against the same dilution of *S. aureus*. The PBS-silver compounds achieved a more significant reduction in viscosity and higher conductivity than the PBS-curcumin compounds at the same weight ratio, but contrary to our expectations, we could not spin PBS-silver compounds with a silver weight ratio exceeding 3% (*w/w*). Given that the conductivity of the PBS-silver compounds was not dependent on the silver content, we concluded that the inability to spin fibers from compounds S5, S7, and S10 reflected their higher content of ash or byproducts from the calcination step, which would favor the formation of agglomerates too large to spin. Overall, the finest fibers were produced from compound C10 and were 12.57 µm in diameter, ~58% finer than pure PBS fibers. The finest silver-loaded fibers were produced from compound S3 and were 16.07 µm in diameter, ~47% finer than pure PBS. The thinnest fibers as a function of weight ratio were obtained from silver compounds, probably reflecting the lower viscosity and higher conductivity than compounds containing the same weight ratio of curcumin. The trends in fiber thickness suggest that S5, S7, and S10 fibers should be even finer than S3 fibers, but a filtration step that removes agglomerates from the melt would be needed to spin such fibers successfully. We conclude that curcumin and silver nanoparticles can both be used as multifunctional additives during melt electrospinning to reduce the viscosity of the PBS melt and increase its electrical conductivity, thus producing narrower fibers with in-built antibacterial activity, suitable for biomedical applications.

## Figures and Tables

**Figure 1 nanomaterials-12-00283-f001:**
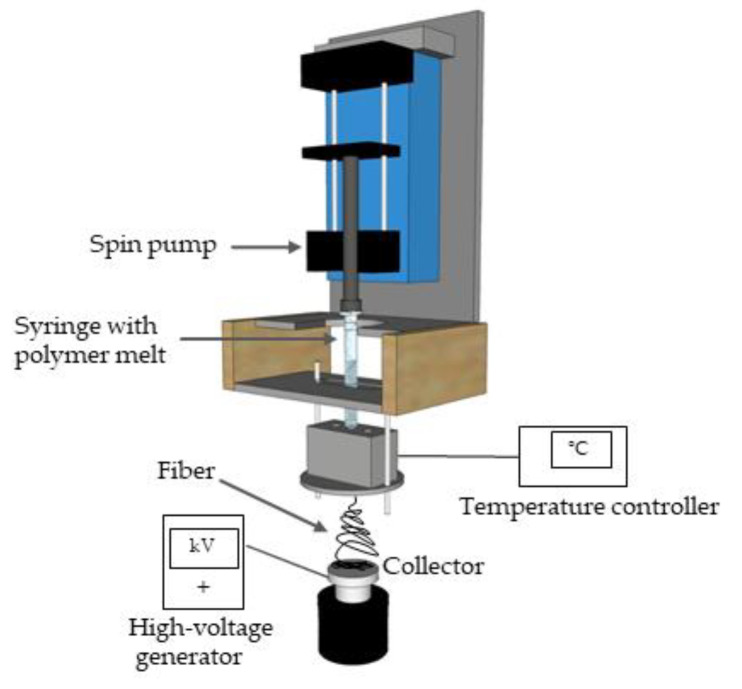
Schematic illustration of the laboratory-scale melt-electrospinning device.

**Figure 2 nanomaterials-12-00283-f002:**
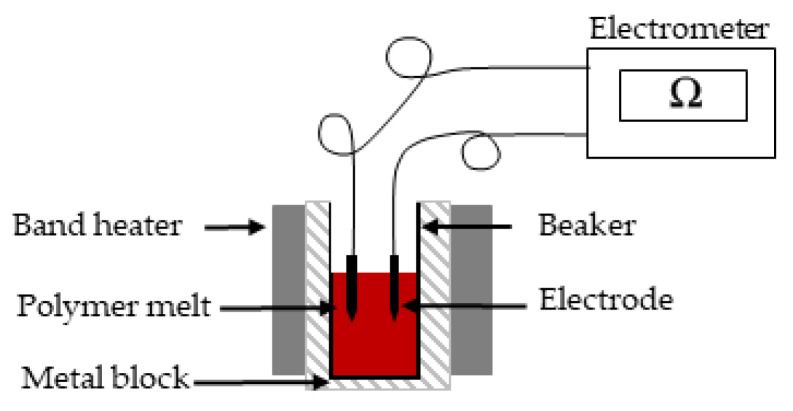
Electrometer apparatus for the analysis of electrical resistance in the PBS melt.

**Figure 3 nanomaterials-12-00283-f003:**
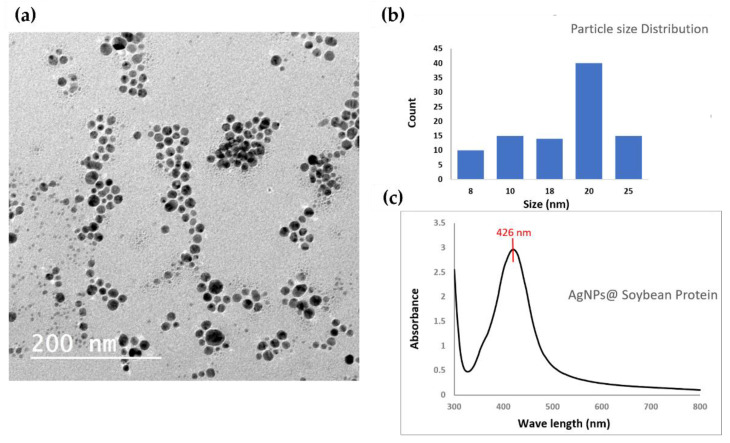
Characterization of silver nanoparticles, (**a**) transmission electron microscope image, (**b**) particle size distribution calculated from TEM image, and (**c**) surface plasmon resonance of AgNPs using UV-Vis spectrophotometer.

**Figure 4 nanomaterials-12-00283-f004:**
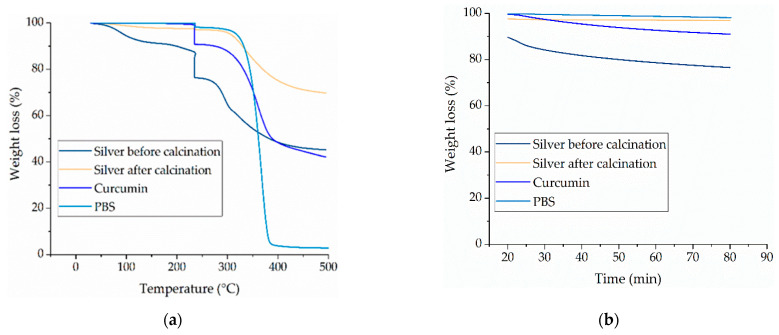
Thermogravimetric analysis (TGA) of the PBS base polymer and pure additives. (**a**) Temperature-dependent degradation profile of PBS, curcumin and silver (before and after calcination). (**b**) Isothermal analysis of PBS, curcumin and silver nanoparticles (before and after calcination) at 235 °C for 60 min.

**Figure 5 nanomaterials-12-00283-f005:**
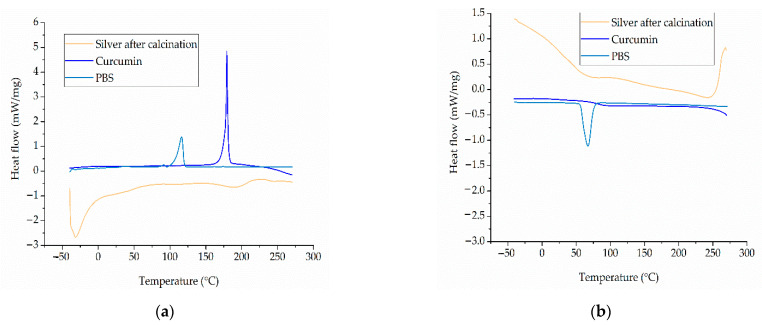
DSC thermograms of PBS, curcumin, and silver. (**a**) Heating cycle. (**b**) Cooling cycle.

**Figure 6 nanomaterials-12-00283-f006:**
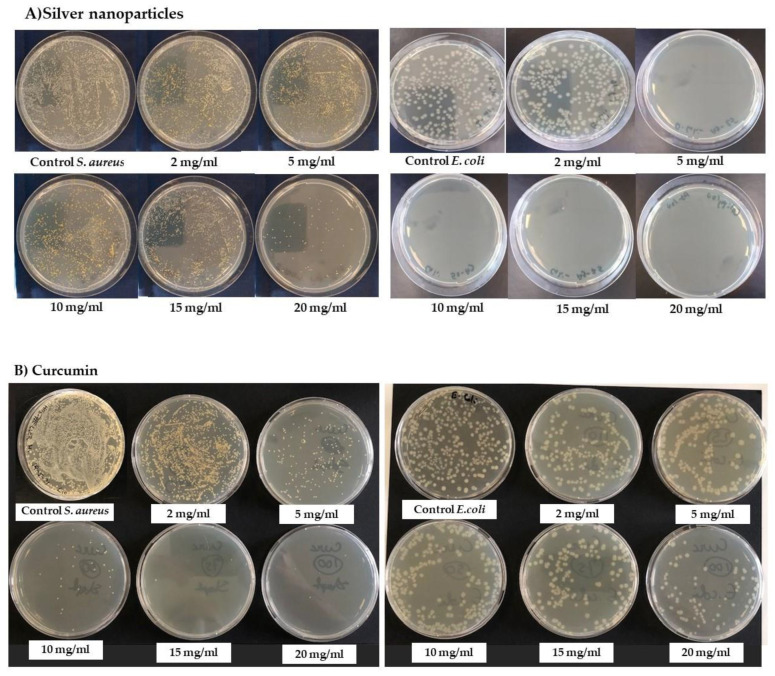
Antibacterial activity assays against Staphylococcus aureus and Escherichia coli. (**A**) Activity of silver nanoparticles. (**B**) Activity of curcumin.

**Figure 7 nanomaterials-12-00283-f007:**
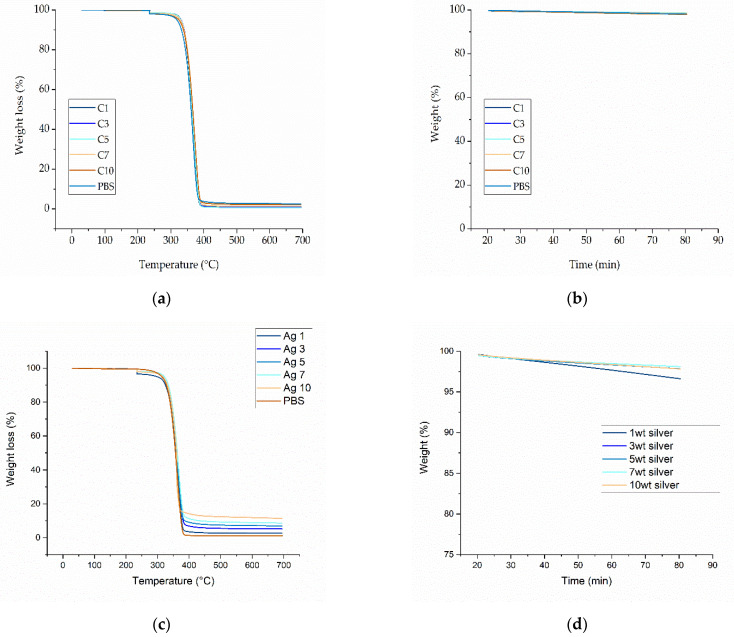
Thermogravimetric analysis (TGA) of the PBS compounds. (**a**) Temperature-dependent degradation profile of PBS compounds with increasing weight percentages of curcumin. (**b**) Isothermal study of PBS compounds with increasing weight percentages of curcumin for 60 min at 235 °C. (**c**) Temperature-dependent degradation profile of PBS compounds with increasing weight percentages of silver. (**d**) Isothermal study of PBS compounds with increasing weight percentages of silver for 60 min at 235 °C.

**Figure 8 nanomaterials-12-00283-f008:**
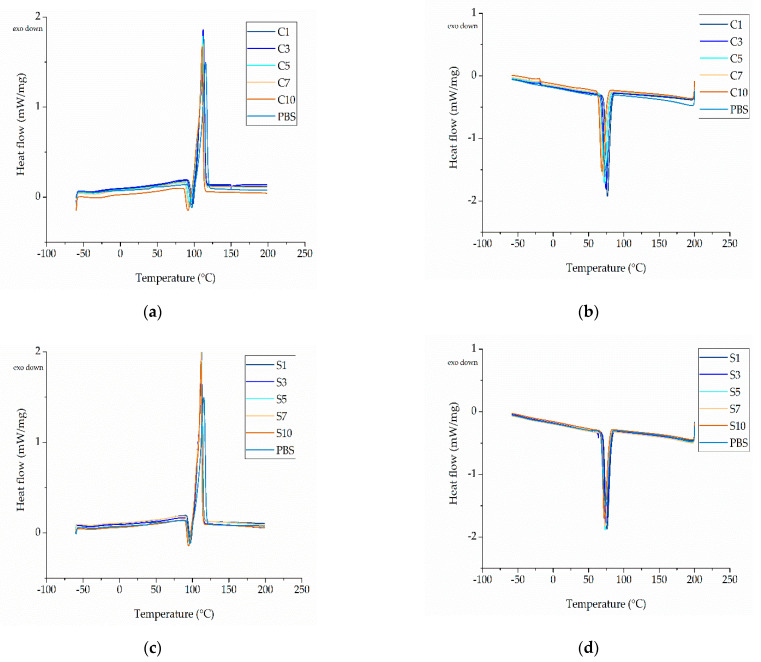
DSC thermogram of PBS compounds. (**a**) Heating cycle of PBS and PBS-curcumin compounds. (**b**) Cooling cycle of PBS and PBS-curcumin compounds. (**c**) Heating cycle of PBS and PBS-silver compounds. (**d**) Cooling cycle of PBS and PBS-silver compounds.

**Figure 9 nanomaterials-12-00283-f009:**
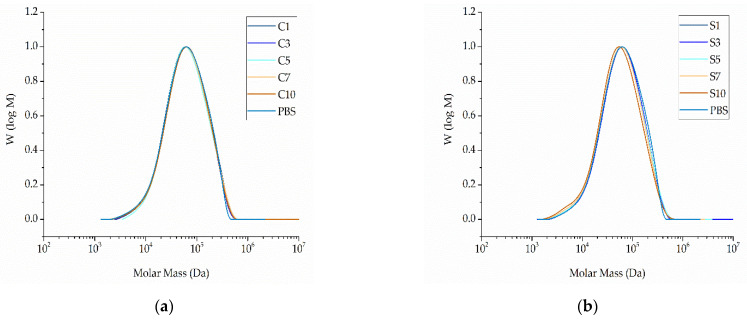
Molar mass distribution curves of PBS and its compounds. (**a**) PBS and compounds containing an increasing weight percentage of curcumin compounded at 200 °C. (**b**) PBS and compounds containing an increasing weight percentage of silver compounded at 200 °C.

**Figure 10 nanomaterials-12-00283-f010:**
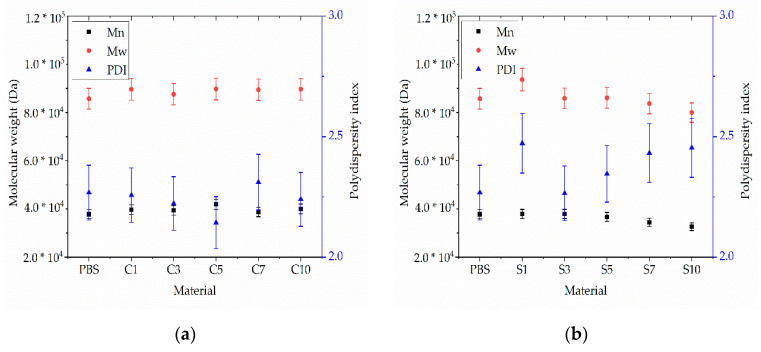
Weight average relative molecular weight (Mw), number average molar mass (Mn) and polydispersity index (PDI) of (**a**) PBS-curcumin compounds and (**b**) PBS-silver compounds. Data are means ± standard deviations (*n* = 3).

**Figure 11 nanomaterials-12-00283-f011:**
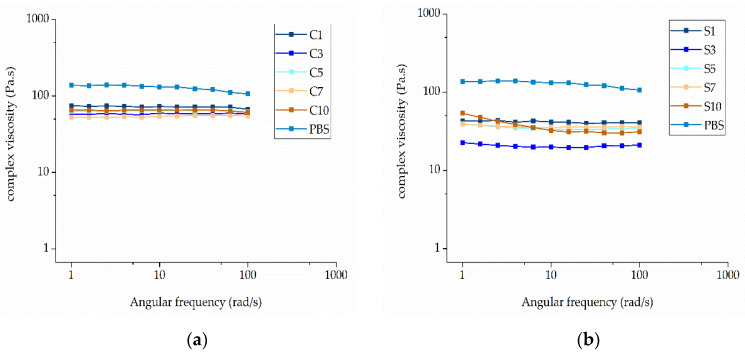
Rheogram showing the complex viscosity of (**a**) PBS-curcumin compounds and (**b**) PBS-silver compounds with increasing angular frequency at 235 °C.

**Figure 12 nanomaterials-12-00283-f012:**
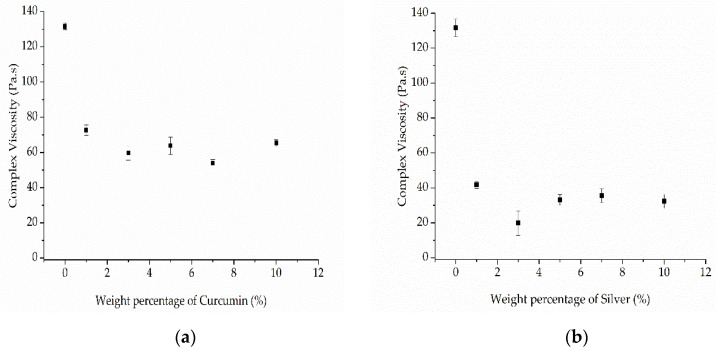
Rheogram showing the complex viscosity of (**a**) PBS-curcumin compounds and (**b**) PBS-silver compounds as a function of temperature at an angular frequency of 10 rad/s. Data are means ± standard deviations (*n* = 3).

**Figure 13 nanomaterials-12-00283-f013:**
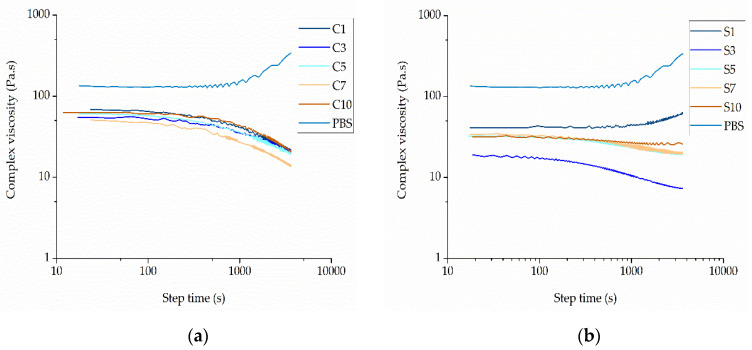
Rheogram showing the complex viscosity of (**a**) PBS-curcumin compounds and (**b**) PBS-silver compounds with increasing step time at 235 °C.

**Figure 14 nanomaterials-12-00283-f014:**
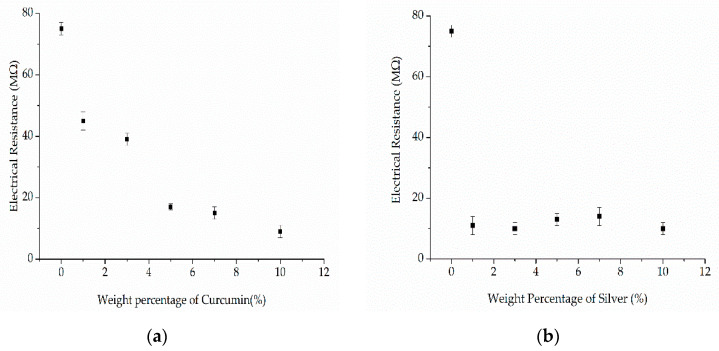
The electrical resistance (MΩ) of (**a**) PBS-curcumin compounds and (**b**) PBS-silver compounds at a polymer melt temperature of 235 °C. Data are means ± standard deviations (*n* = 3).

**Figure 15 nanomaterials-12-00283-f015:**
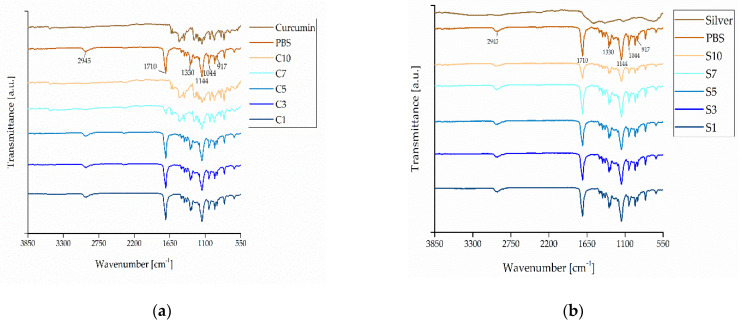
FTIR spectra of PBS, the additives, and the resulting compounds. (**a**) PBS, curcumin and PBS-curcumin compounds. (**b**) PBS, silver and PBS-silver compounds.

**Figure 16 nanomaterials-12-00283-f016:**
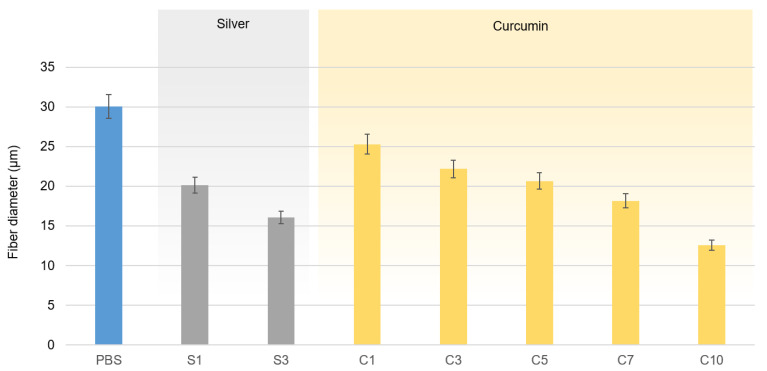
Diameter of PBS and PBS compound fibers produced by melt-electrospinning using a single-nozzle laboratory device at 235 °C and 60 kV, with a nozzle-to-collector distance of 10 cm. Data are means ± standard deviations (*n* = 100).

**Figure 17 nanomaterials-12-00283-f017:**
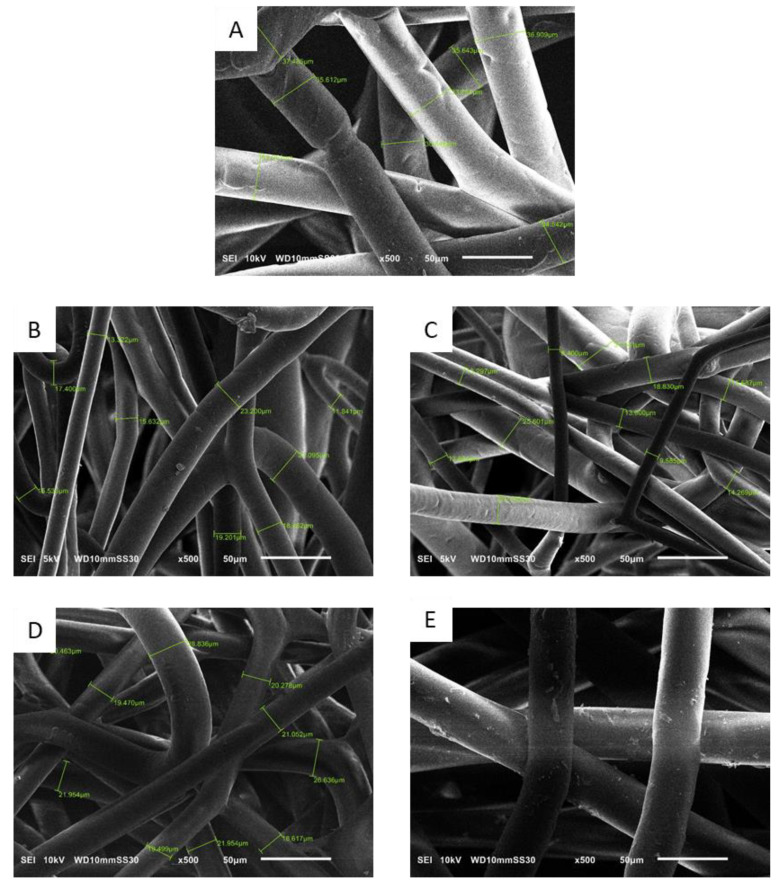
Scanning electron micrographs of PBS melt-electrospun microfibers. (**A**) Pure PBS. (**B**) S1 fibers. (**C**) S3 fibers. (**D**) C1 fibers. (**E**) C3 fibers.

**Figure 18 nanomaterials-12-00283-f018:**
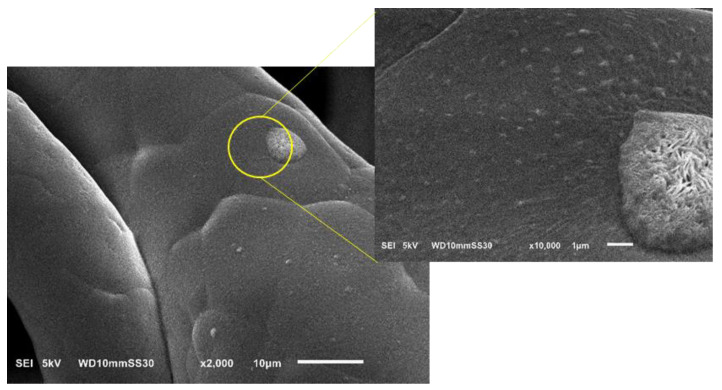
Scanning electron micrographs of PBS melt-electrospun microfibers containing 3% silver nanoparticles (S3). The inset image shows the silver particles homogeneously distributed on the fiber.

**Figure 19 nanomaterials-12-00283-f019:**
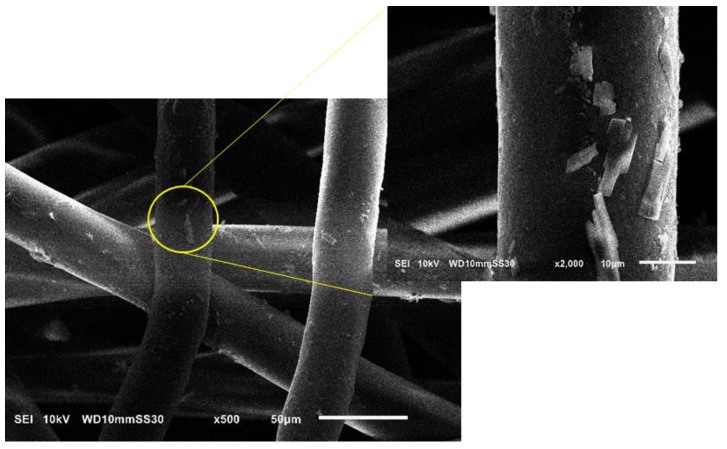
Scanning electron micrographs of PBS melt-electrospun microfibers containing 3% curcumin (C3). The inset image shows curcumin flaks distributed on the fibers.

**Figure 20 nanomaterials-12-00283-f020:**
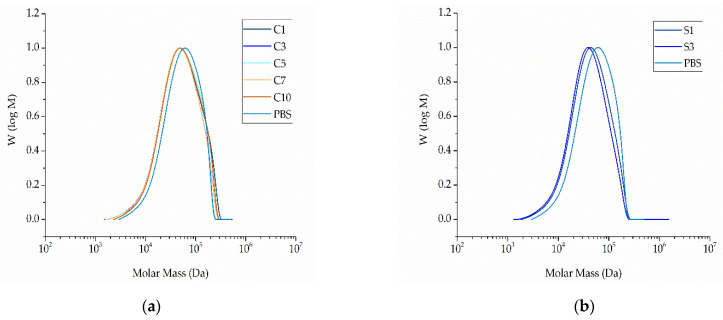
Molar mass distribution curves. (**a**) Fibers composed of PBS compounds with increasing weight percentages of curcumin. (**b**) Fibers composed of PBS compounds with increasing weight percentages of silver. All fibers were prepared by melt electrospinning at 235 °C and 60 kV, with a nozzle-to-collector distance of 10 cm and a throughput of 0.1 mL/min.

**Figure 21 nanomaterials-12-00283-f021:**
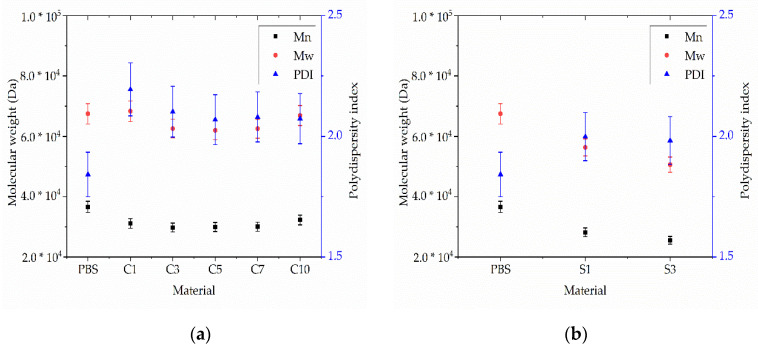
Weight average relative molecular weight (Mw), number average molar mass (Mn) and polydispersity index (PDI) of (**a**) PBS-curcumin compound fibers and (**b**) PBS-silver compound fibers. Data are means ± standard deviations (*n* = 3).

**Figure 22 nanomaterials-12-00283-f022:**
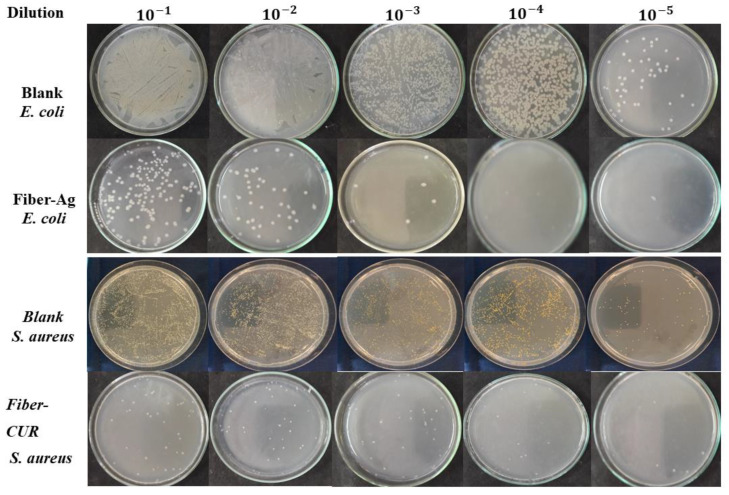
Antibacterial activity of PBS fibers loaded with curcumin or silver nanoparticles.

**Table 1 nanomaterials-12-00283-t001:** Chemical structures and melting points of polybutylene succinate (PBS) and curcumin.

Material	Chemical Structure	Melting Point (°C)
PBS	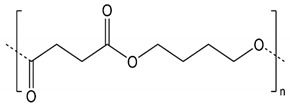	115 °C
Curcumin	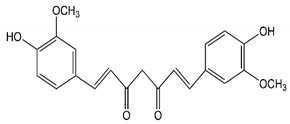	175 °C

**Table 2 nanomaterials-12-00283-t002:** Compound abbreviations according to the dye and weight percentage.

Compound Abbreviation	Additive	Percentage of Additive (*w/w*)
C1	Curcumin	1
C3	Curcumin	3
C5	Curcumin	5
C7	Curcumin	7
C10	Curcumin	10
S1	Silver nanoparticles	1
S3	Silver nanoparticles	3
S5	Silver nanoparticles	5
S7	Silver nanoparticles	7
S10	Silver nanoparticles	10

**Table 3 nanomaterials-12-00283-t003:** Initial degradation temperatures and weight shares at 235 °C for PBS, curcumin and silver.

Material	Residual Weight before Isothermal (235 °C) (%)	Residual Weight after Isothermal (%)	50% Weight Loss Temperature
PBS	99.50%	97.76%	398.00 °C
Curcumin	99.14%	91.00%	389.10 °C
Silver before calcination	89.67%	76.53%	383.37 °C
Silver after calcination	97.58%	96.97%	-

**Table 4 nanomaterials-12-00283-t004:** Initial degradation temperatures and weight shares at 235 °C for PBS as well as PBS-curcumin and PBS-silver compounds.

Material	Residual Weight until 235 °C (%)	Residual Weight after Isothermal (%)	50% Weight Loss Temperature
PBS	99.85%	98,15%	356.00 °C
C1	99.64%	98.55%	364.34 °C
C3	99.67%	98.57%	365.83 °C
C5	99.70%	98.69%	364.60 °C
C7	99.67%	98.48%	366.75 °C
C10	99.66%	98.01%	366.51 °C
S1	99.62%	96.63%	358.92 °C
S3	99.56%	97.83%	361.07 °C
S5	99.45%	97.85%	360.44 °C
S7	99.46%	98.11%	362.78 °C
S10	98.74%	98.19%	360.66 °C

**Table 5 nanomaterials-12-00283-t005:** The melting temperature (Tm), re-cooling temperature (Trc), crystallization temperature (Tc) and degree of crystallinity (Xc) of PBS and its compounds. Data are means ± standard deviations (*n* = 3).

Material	Tm1 (°C)	Tm2 (°C)	Trc (°C)	Tc (°C)	Xc (%)
PBS		112.41 ± 1	96.95 ± 2	74.41 ± 1	55.42 ± 1
C1	107.89 ± 2	112.46 ± 1	98.25 ± 1	74.78 ± 1	55.44 ± 1
C3	106.89 ± 2	112.15 ± 1	96.26 ± 2	73.24 ± 1	56.41 ± 1
C5	105.88 ± 1	111.71 ± 1	94.29 ± 3	71.55 ± 3	57.37 ± 2
C7	105.54 ± 1	111.19 ± 1	92.63 ± 2	69.62 ± 3	58.60 ± 2
C10	104.53 ± 1	110.73 ± 2	91.50 ± 3	68.07 ± 3	59.24 ± 2
S1	-	112.16 ± 1	98.15 ± 2	76.04 ± 2	55.48 ± 1
S3	-	112.12 ± 1	99.26 ± 2	78.36 ± 3	55.63 ± 1
S5	-	111.95 ± 1	95.57 ± 3	73.42 ± 1	55.80 ± 1
S7	-	111.87 ± 1	96.07 ± 2	73.41 ± 1	56.06 ± 2
S10	-	111.60 ± 1	95.26 ± 3	72.36 ± 1	55.08 ± 1

## Data Availability

The datasets used and/or analyzed during this study are available from the corresponding author on reasonable request.

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
