# Peer review of "Curcumin and Silver Doping Enhance the Spinnability and Antibacterial Activity of Melt-Electrospun Polybutylene Succinate Fibers"

_nanomaterials, 2022, doi:10.3390/nano12020283_

Round 1

Reviewer 1 Report

In this paper, the effect of the addition of curcumin and silver nanoparticles on the melt-electrospinnability of PBS and the antibacterial activity of the PBS-based composite fibers were studied. In general, this paper was well written and the proposed strategy could provide instructions for the development of high-performance PBS-based composite fiber materials. However, the manuscript should be revised substantially with the following comments to be considered before being accepted by Nanomaterials.

  1. “doping”is used in the tile of this paper. However, the curcumin or silver molecules has not been integrated into PBS polymer backbone to form doping structure. I think that the incorporation or addition may be more proper to describe the relationship between the additives and PBS.
  2. There are too many figure in the manuscript. The figures demonstrating similar information should be merged into one figure. For example, TGA results should be compiled and shown in one or two figures. The unimportant figures can be transferred to supporting information.
  3. In the introduction section, previous work about antibacterial nanofibers containing silver (i.e. Appl. Polym. Sci. 2018, 135, 46238) should be supplemented and commented.
  4. The diameter of silver nanoparticles should be provided. Additionally, the size of added silver nanoparticles may influence the electrical conductivity and viscosity of PBS melt, resulting in PBS fiber with difference morphology. The author can provide the related discussions about the relationship between particle size and fiber morphology.  

Author Response

Comment

Rebuttal

doping”is used in the tile of this paper. However, the curcumin or silver molecules has not been integrated into PBS polymer backbone to form doping structure. I think that the incorporation or addition may be more proper to describe the relationship between the additives and PBS.

We agree with the reviewer that incorporation or addition are also suitable in the title. However, we believe doping also means the addition of particles to a system and grafting is the term more generally used for incorporating additives in the polymer backbone. Also, doping is a common term used in electronics when adding impurities to acquire specific functions. Some examples in the literature are shown here:

https://doi.org/10.1016/S0379-6779(01)00510-0

https://doi.org/10.1063/1.1343854.

Therefore, we would prefer to leave the title unchanged.

There are too many figures in the manuscript. The figures demonstrating similar information should be merged into one figure. For example, TGA results should be compiled and shown in one or two figures. The unimportant figures can be transferred to supporting information.

The consolidation of graphs to reduce the number of figures unfortunately made the information in the graphs difficult to understand, so we have reverted to the original number of figures.

In the introduction section, previous work about antibacterial nanofibers containing silver (i.e. Appl. Polym. Sci. 2018, 135, 46238) should be supplemented and commented.

We thank the reviewer for highlighting this interesting article and have cited it as requested (lines 74 and 85-89).

The diameter of silver nanoparticles should be provided. Additionally, the size of added silver nanoparticles may influence the electrical conductivity and viscosity of PBS melt, resulting in PBS fiber with difference morphology. The author can provide the related discussions about the relationship between particle size and fiber morphology.

We have added some discussion of the relationship between particle size and fiber morphology between lines 597-600. We also characterized the (as prepared) silver nanoparticles by TEM and UV-Vis and added a reference to our previous work concerning the synthesis of silver nanoparticles. The average particle diameter distribution was calculated from TEM images and presented in an additional figure.  Regarding the effect of particle size on fiber diameter, we used the polyol method to prepare the nanoparticles, which limited our control over the particle size. We used only one sample of silver nanoparticles with an average particle diameter of 20 nm. 

Reviewer 2 Report

The scientific paper is normal and does not contain any kind of innovation.  There are two important points. The first is that the diameter of the obtained fibers is in the form of microns, so how is the title of obtaining nanofibers, and the developed fibers were 12.57 microns (12570 nanometers) in diameter?
Secondly, there are no comparisons between other additives that were used in previous works and the additives in this work. 

The resolution of all figures in the manuscript have very poor clarity.

The decision to publish depends on the opinion of the editor, because there are better scientific articles than this article, and it was rejected by the editorial office.

Author Response

Comment

Rebuttal

The scientific paper is normal and does not contain any kind of innovation.  There are two important points. The first is that the diameter of the obtained fibers is in the form of microns, so how is the title of obtaining nanofibers, and the developed fibers were 12.57 microns (12570 nanometers) in diameter?

We acknowledge the reviewer’s candid criticism but we clearly state in the manuscript that our experiments “provide the basis to develop environmentally benign antibacterial melt-electrospun PBS fibers for biomedical applications” by testing the effect of two additives. It is not entirely clear what the reviewer expects in terms of innovation, but the objectives of the work and experimental approach are clearly described: “We investigated the preparation of PBS microfibers incorporating different weight percentages of two multifunctional additives … using a single-nozzle laboratory-scale device” to determine their influence on the properties of the resulting fibers.

With regard to the title, it is “Curcumin and silver doping enhance the spinnability and antibacterial activity of melt-electrospun polybutylene succinate fibers”. We do not mention nanofibers in the title, and as mentioned above, we state in the abstract that “We investigated the preparation of PBS microfibers….”

Secondly, there are no comparisons between other additives that were used in previous works and the additives in this work.

We have revised the manuscript to provide more comparisons as suggested, and the new information can be found on lines 31-320, 655–658.

The resolution of all figures in the manuscript has very poor clarity.

We prepared all images using Origin and they are saved as .tiff files with a resolution of 300 dpi, which is the resolution asked for in most journals.

The decision to publish depends on the opinion of the editor, because there are better scientific articles than this article, and it was rejected by the editorial office.

We refute this comment entirely. Solution electrospinning is a well-studied process whereas melt-electrospinning is a greener process that is gaining more attention now it is possible to generate fibers with a low diameter. The melt electrospinning of PBS has not been studied in detail, despite the potential of this material. We use multifunctional additives to optimize the melt electrospinning process and introduce antibacterial properties. We present our data in a logical manner, first addressing the properties of PBS and the additives individually and then as compounds before analyzing the resulting fibers, and finally discussing the relationship between the composition of the compounds and the fiber properties.  Again, we appreciate the reviewer’s candor but we feel the criticism is somewhat unconstructive and lacks objectivity. We also note that this reviewer’s opinion is not shared by the other reviewers.

Reviewer 3 Report

This paper reports an interesting work on the development of polybutylene succinate fibers (PBS) loaded with curcumin and silver nanoparticles. The fibers were obtained by electrospinning and the incorporation of the additives curcumin and silver nanoparticles was able to reduce the fiber diameter, as well as to add antibacterial capacity to the fibers. This work is highlighted by the results obtained, especially on the fiber diameter, and the biological assays conducted in terms of antibacterial capacity. Overall, the document is well written, though Results and Discussion should be better organised and explained. The section division chosen is not easily perceptive. There are some important issues and typos to be taken in consideration, before any publication in Nanomaterials.

Typos:

P6 L225: E. coli … S. aureus (in Italic)

P8 L291: strain

P9 L306: “The 50” is bold?

P9 L314: The 50% weight loss temperatures

P14 L409: respectively), as shown in Figure 9.

P15 L434: have also low molecular

Figure 5: In figure “A) Silver” space missing.

Formatting:

P5 L190-193: Text is not justified.

P6 L236-238: Text is not justified.

Issues:

  1. The Results and Discussion section is somehow confusing, specially section 3.2 and 3.3. While reading, it is not easily understanding the different between the Compound analysis and the Fiber analysis, since the results outcome is mainly the same. A brief introduction is highly required in the beginning of each section, and not jumping directly to data.
  2. Concerning the previous issue, special attention should be given to the legends. For instance, Figure 9 and Figure 20 have exactly the same legend. What is different between both analysis? This is also confusing when referring to figures in text (P22 L577: “in Figure 20Figure 9”).
  3. Antibacterial activity is shown with photos of the petri dishes. Why did authors not show the results in terms of colonies counted? Was it replicated?

Author Response

Comment

Rebuttal

This paper reports an interesting work on the development of polybutylene succinate fibers (PBS) loaded with curcumin and silver nanoparticles. The fibers were obtained by electrospinning and the incorporation of the additives curcumin and silver nanoparticles was able to reduce the fiber diameter, as well as to add antibacterial capacity to the fibers. This work is highlighted by the results obtained, especially on the fiber diameter, and the biological assays conducted in terms of antibacterial capacity. Overall, the document is well written, though Results and Discussion should be better organised and explained. The section division chosen is not easily perceptive. There are some important issues and typos to be taken in consideration, before any publication in Nanomaterials.

We thank the reviewer for these constructive comments. We have organized the results and discussion in order to first describe the properties of PBS and the additives individually and then as compounds before analyzing the resulting fibers, and finally, discussing the relationship between the composition of the compounds and the fiber properties.

Typos:

P6 L225: E. coli … S. aureus (in Italic)

P8 L291: strain

P9 L306: “The 50” is bold?

P9 L314: The 50% weight loss temperatures

P14 L409: respectively), as shown in Figure 9.

P15 L434: have also low molecular

Figure 5: In figure “A) Silver” space missing

 We thank the reviewer for highlighting the errors and have made the requested corrections.

Formatting:

P5 L190-193: Text is not justified.

P6 L236-238: Text is not justified.

We thank the reviewer for highlighting the errors and have made the requested corrections.

The Results and Discussion section is somehow confusing, specially section 3.2 and 3.3. While reading, it is not easily understanding the different between the Compound analysis and the Fiber analysis, since the results outcome is mainly the same. A brief introduction is highly required in the beginning of each section, and not jumping directly to data.

We thank the reviewer for these constructive comments. We have added a brief instruction to each of the analysis sections to make the objective of each subsection clear to the reader.

Concerning the previous issue, special attention should be given to the legends. For instance, Figure 9 and Figure 20 have exactly the same legend. What is different between both analysis? This is also confusing when referring to figures in text (P22 L577: “in Figure 20Figure 9”).

We thank the reviewer for these constructive comments and are happy to answer this. Figure 9 represents the GPC analysis of the compounds, whereas Figure 20 represents the GPC analysis of the fibers. The compounding was carried out at a temperature of 200°C, below the processing temperature of 230°C. Furthermore, biopolymers generally show degradation when being heat-processed several times. That’s why we chose to analyse the degradation of both, the compounds and the fibers.

In L577, there was a typographical error and we have corrected it now. 

Antibacterial activity is shown with photos of the petri dishes. Why did authors not show the results in terms of colonies counted? Was it replicated?

Thanks for the comment. We thought that the photos would be more representative and indicative since we have several tables in the manuscript. However, a table with colony count would also work. All the biological experiments were done in triplicates. We added a note to the experimental part.

Reviewer 4 Report

The manuscript present the development of PBS composite fibers via melt electrospinning. It contains a vast amount of information and characterization results using a variety of methods. It is very interesting and recommended for publication. However, before that, the following issues should be addressed.

  • More details about the silver nanoparticles are needed. The authors mention that neat nanoparticles consist of an organic part, the existence of which is not obvious to the reader. The authors have a reference to a previous study, but in my opinion this is not enough and some characterization of those nanoparticles is need as information in this manuscript.
  • Also, the authors mention that silver nanoparticles show a melting range before 200 ο It is surprising for inorganic materials (calcinated nanoparticles) to melt at those low temperatures. Which is the origin of this melting behavior? Which is the reported melting temperature range, since in Figure 4, it is not obvious that a melting behavior occurs?

Considering the aforementioned issues, more analysis and more characterization details for such material is needed, before reaching conclusions.

  • Why S10 presents so different decomposition (50% weight loss) temperature compared to the other composites? What is the reason for this? The authors should try to discuss and explain it. However, I suggest a verification experiment, if this haven’t been already performed, in order to diminish the possibility of an experimental error.

Author Response

Comment

Rebuttal

More details about the silver nanoparticles are needed. The authors mention that neat nanoparticles consist of an organic part, the existence of which is not obvious to the reader. The authors have a reference to a previous study, but in my opinion, this is not enough and some characterization of those nanoparticles is need as information in this manuscript.

We thank the reviewer for these constructive comments. We have added more details about silver nanoparticles to the methods section. Soybean protein was used to prepare silver nanoparticles in a solid-state reaction. In addition, the formation of nanoparticles was confirmed by TEM. We have added a figure to show the particle size distribution.

Also, the authors mention that silver nanoparticles show a melting range before 200 ο It is surprising for inorganic materials (calcinated nanoparticles) to melt at those low temperatures. Which is the origin of this melting behavior? Which is the reported melting temperature range, since in Figure 4, it is not obvious that a melting behavior occurs?

We observed a very broad peak from ~ 50–250 degrees and we believe this reflects the melting of organic components (soybean protein) present in the silver extract. We have modified the text to discuss the results in more detail (lines 318–324).

Considering the aforementioned issues, more analysis and more characterization details for such material is needed, before reaching conclusions.

We have included information on synthesis of the silver nanoparticle (128-136) and characterized the (as prepared) silver nanoparticles by TEM and UV-Vis and added a reference to our previous work concerning the synthesis of silver nanoparticles (257-270). The average particle diameter distribution was calculated from TEM images and presented in an additional figure.  Regarding the effect of particle size on fiber diameter, we used the polyol method to prepare the nanoparticles, which limited our control over the particle size. We used only one sample of silver nanoparticles with an average particle diameter of 20 nm. 

Why S10 presents so different decomposition (50% weight loss) temperature compared to the other composites? What is the reason for this? The authors should try to discuss and explain it. However, I suggest a verification experiment, if this haven’t been already performed, in order to diminish the possibility of an experimental error.

We performed the verification experiment twice to check the degradation temperature of PBS and S10. However, we now see that the values are not as low as we initially reported, suggesting an error during the setup of the TGA experiment. Our new experiments yielded results much closer to the value of the other compounds. We have amended the manuscript accordingly.

Round 2

Reviewer 1 Report

The authors have addressed the original comments and I believe the manuscript is now acceptable for publication.

Author Response

Thank you very much for your valuable input. 

Reviewer 3 Report

I am glad to see the improvements made to the previous version. I think this version can be accepted for Nanomaterials publication.

However, there is still one important issue I would like to see improved. In the Section 3 (Results and Discussion), the point is not to have a sentence in the beggining of each sub-section, only changing the title wording. The meaning is to have a brief introduction after P7 L259, stating how section 3 is organized (material analysis, compound analysis and fiber analysis) and which is the purpose of each sub-section. 

I believe this will improve the impact on reading the paper.

Author Response

Thank you very much for your valuable input. We have added a short explanation about the structure of section 3 and also pointed out the sense of each sub section. The described changes can be found on page 7, line 260-268.

Reviewer 4 Report

No further comments. The authors made all needed improvements. 

Author Response

(The authors gave the same response as above.)
